# Mitochondrial Dysfunction in the Transition from NASH to HCC

**DOI:** 10.3390/metabo9100233

**Published:** 2019-10-16

**Authors:** Mélissa Léveillé, Jennifer L. Estall

**Affiliations:** 1Institut de Recherches Cliniques de Montréal (IRCM), Montreal, Quebec, QC H2W 1R7, Canada; Melissa.Leveille@ircm.qc.ca; 2Faculty of Medicine, University of Montreal, Montreal, Quebec, QC H3G 2M1, Canada; 3Division of Experimental Medicine, McGill University, Montreal, Quebec, QC H4A 3J1, Canada

**Keywords:** mitochondria, metabolism, liver, NAFLD, NASH, HCC

## Abstract

The liver constantly adapts to meet energy requirements of the whole body. Despite its remarkable adaptative capacity, prolonged exposure of liver cells to harmful environmental cues (such as diets rich in fat, sugar, and cholesterol) results in the development of chronic liver diseases (including non-alcoholic fatty liver disease (NAFLD) and non-alcoholic steatohepatitis (NASH)) that can progress to hepatocellular carcinoma (HCC). The pathogenesis of these diseases is extremely complex, multifactorial, and poorly understood. Emerging evidence suggests that mitochondrial dysfunction or maladaptation contributes to detrimental effects on hepatocyte bioenergetics, reactive oxygen species (ROS) homeostasis, endoplasmic reticulum (ER) stress, inflammation, and cell death leading to NASH and HCC. The present review highlights the potential contribution of altered mitochondria function to NASH-related HCC and discusses how agents targeting this organelle could provide interesting treatment strategies for these diseases.

## 1. Introduction

The global epidemic of obesity correlates with the rising prevalence of the metabolic syndrome, a cluster of metabolic abnormalities increasing the risk of cardiovascular disease and type 2 diabetes mellitus (T2DM) [1]. Nonalcoholic fatty liver disease (NAFLD) is the hepatic manifestation of the metabolic syndrome and is now the most common chronic liver disease in Western countries, affecting approximately 30% of the general population [2]. Importantly, about 70% of obese patients with T2DM have NAFLD [3,4,5]. While most studies report that NAFLD is significantly more prevalent in men [6], following menopause, this relationship reverses and women become equally (if not more) susceptible to NAFLD [7]. NAFLD represents a wide spectrum of liver diseases ranging from simple steatosis (triacylglycerol infiltration in >5% of hepatocytes) to a more severe necro-inflammatory form called nonalcoholic steatohepatitis (NASH) resulting in fibrosis [8,9,10]. Hepatocellular carcinoma (HCC) is more prevalent within the setting of NASH than NAFLD (2.4%–12.8% versus 0%–3%) [11], but the precise mechanism(s) that give rise to cancer within this altered metabolic environment remain unknown.

There was a 9% annual increase in HCC associated with NAFLD between 2004 and 2009 [12] and a study surveying 18 million subjects from 2002 to 2008 found NAFLD/NASH to be the leading co-morbidity in 38.2% of HCC cases [13]. Hepatocellular carcinoma is the fifth leading cause of new cancer cases, the second leading cause of cancer death in men and sixth in women, worldwide [14]. Liver resection, orthotopic transplantation, and systemic chemotherapy are the only currently accepted therapeutic options for HCC [15]. Unfortunately, HCC is minimally responsive to chemotherapy and most cases present only at a late stage, often making patients unsuitable candidates for transplantation [16,17,18]. Emerging data suggest that NASH and metabolic syndrome have a higher proportion of HCCs manifesting in the absence of cirrhosis [19,20,21,22], strikingly different compared to HCC of other etiologies, which require a cirrhotic step. Consequently, NASH-related HCC is less likely to be diagnosed by surveillance compared to HCC secondary to viral hepatitis [22]. Obesity-associated NASH is currently the third leading cause for liver transplantation and may surpass hepatitis C and alcohol consumption as the most common indication for liver transplantation in the developed world [23]. A better understanding of the molecular mechanisms underlying the relationship between hepatic metabolism and carcinogenesis is needed and would greatly contribute to the development of new therapeutic strategies.

The liver is a central organ responsible for carbohydrate, lipid, and protein metabolism. It is also one of the richest organs in terms of number and density of mitochondria, serving as a critical site for multiple metabolic pathways including β-oxidation, tricarboxylic acid (TCA) cycle, ketogenesis, respiratory activity, and adenosine triphosphate (ATP) synthesis, providing metabolic fuels for itself and the rest of the body [24]. Given the importance of mitochondria to hepatic function, it is not surprising that evidence points toward inadequate mitochondrial adaptation as a likely central player in the pathological transition from NASH to HCC [3,25,26,27]. Metabolic reprogramming is a key feature of hepatocellular carcinoma and mitochondria defects are well documented in HCC [28,29]. What remains a mystery is how and why mitochondria fail to adapt to metabolic challenges and whether targeting this aspect of the disease would prevent or reverse the progression to cancer. Deciphering if there is a role for mitochondria in the progression of NASH to HCC is also critical and offers a new horizon for therapeutic options. Herein, we review what is currently known about altered hepatic mitochondrial function in NASH and related liver cancer.

## 2. Mitochondrial Dysfunction in Non-Alcoholic Steatohepatitis (NASH)

### 2.1. Mitochondrial Adaptation and Flexibility

Increased mitochondrial mass and biogenesis in liver tissue is reported in NAFLD and NASH patients [27], supporting the theory that mitochondria are highly adaptive and increase in number and capacity in environments of substrate excess. However, in contrast with the often-increased mitochondrial activity (i.e., β-oxidation, mitochondrial respiration, TCA cycle, and ketogenesis) observed in early stages of NAFLD in response to hepatic insulin resistance and free fatty acid overload [30,31,32,33,34,35], mice and humans with NASH exhibit blunted ketogenesis [31,36], lower maximal respiration, mitochondrial uncoupling, and leakage [31,37]. This more severe stage of the disease is also associated with an overactive mitochondrial TCA cycle potentially to meet the high energy demand [30]. Impaired coupling of substrate oxidation and ATP production is in line with previous findings in obese subjects and mice [38,39,40,41,42,43]. This suggests that mitochondrial adaptation and flexibility become compromised once (or possibly just before) NAFLD progresses to steatohepatitis. Adaptation or remodeling of mitochondrial energetics in the pathogenesis of simple steatosis to NASH is elegantly reviewed by Sunny and colleagues [3], and we will focus specifically on aspects of mitochondrial dysfunction that seem to be important for the NASH to HCC transition (Figure 1).

### 2.2. The Delicate Balance of Mitochondrial Reactive Oxygen Species

The imbalance of respiratory complex activity during the development of NAFLD and its progression to NASH leads to increased mitochondrial reactive oxygen species (ROS), causing oxidative mitochondrial DNA (mtDNA) damage, mitochondria structural abnormalities, and lipid peroxidation [37,44,45]. In line with this, increased serum markers of oxidation such as malondialdehyde are detected in NAFLD subjects, while antioxidants coenzyme Q10 and CuZn-superoxide dismutase are reduced [46]. Mitochondrial ROS (mtROS) and lipid peroxidation also trigger proinflammatory cytokines such as interleukin 6 (IL-6), tumor necrosis factor α (TNF-α), interleukin 1 beta (IL-1β), which are critical mediators of inflammation in NASH [47,48,49,50]. This is observed in rodent models of NASH, as substantial amounts of lipid peroxidation and proinflammatory cytokines are detected in mice and rats with steatohepatitis caused by NASH-promoting diets [51,52]. Exogenous lipid delivery or high-fat feeding in mice also upregulates mitochondrial oxidative metabolism, resulting in increased oxidative stress and inflammation [53].

Mitochondrial-derived ROS activate mitogen-activated protein kinases (MAPKs) including c-Jun N-terminal kinase (JNK) [54]. Phosphorylated JNK translocates to the mitochondria, where it binds to scaffold proteins inhibiting the mitochondrial respiratory chain and further increasing ROS production [54,55,56]. In animal models, sustained JNK activation mediates liver injury, but blocking translocation of phosphorylated JNK to the mitochondria can prevent the development of fatty liver in mice fed a high-fat diet [55]. Mitochondrial-derived ROS also activate AMP-activated protein kinase (AMPK) secondary to redox changes and mitochondrial ATP production [57]. H_2_O_2_ produced by mitochondria induces antioxidant enzyme expression through AMPK-mediated activation of nuclear factor 2 (NRF2), and consistently, NRF2 knockout mice on a high-fat diet develop more severe NASH associated with decreased antioxidant response, reduced β-oxidation genes, and increased lipogenic genes [58]. AMPK activators partially restore fatty acid oxidation in primary hepatocytes, consistent with a study showing that liver-specific activation of AMPK in mice protects against liver steatosis [59]. Taken together, mitochondrial functional status and ROS generation directly impact AMPK and JNK signaling, which both have important roles in maintaining liver metabolic function and health.

AMPK also activates the peroxisome proliferator-activated receptor gamma coactivator 1-alpha (PGC-1α), which is required for mitochondrial adaptive responses to oxidative stress [60]. PGC-1α is a transcriptional coactivator coordinating activity of several transcription factors important for mitochondrial biogenesis and function including, but not limited to, peroxisome proliferator-activated receptors (PPARs), nuclear respiratory factors (NRF-1 and -2), and estrogen-related receptors (ERRs). Decreased mitochondrial biogenesis is associated with a reduction of PGC-1α, transcription factor A, mitochondrial (TFAM), or PPARs in fatty livers [61,62,63,64]. Interestingly, PGC-1α coactivates estrogen receptor-alpha (ERα) in a ligand-dependent manner to increase superoxide dismutase 2 (SOD2) and glutathione peroxidase 1 (GPX1), enhancing ROS scavenging and reducing oxidative damage [52]. Besse-Patin et al. showed that female mice express higher levels of PGC-1α in liver tissue compared to male mice and are more susceptible to reduced PGC-1α in obesity, in line with evidence suggesting dependence of estrogen’s antioxidative properties on coactivator activity [52]. It has not yet been investigated whether protection of liver cells from mitochondria-induced ROS via PGC-1α:ERα contributes to sex-dependent differences in human NASH pathogenesis, where pre-menopausal women are more resistant. Although, it is intriguing to note that PGC-1α and mitochondrial mass/function are known to decrease with age [65,66,67,68,69,70,71], and combined with the drop in estrogen during menopause, this may explain why the incidence of NAFLD and NASH in women approaches (and may surpass) that of men after menopause [7,72].

In line with increased liver PGC-1α being beneficial in NAFLD, liver-specific overexpression of PGC-1α in Sprague Dawley male rats increases fatty acid oxidation and tricarboxylic acid (TCA) cycle activity in isolated liver mitochondria, and this was coupled with reduced hepatic and plasma triglycerides [73]. While hepatic PGC-1α is known to increase in a fasting liver to promote lipid catabolism, it is interesting to note that it also increases following acute exercise training in rodents [74,75,76,77]. McCoin et al. investigated whether liver-specific hemizygous disruption of PGC-1α had an impact on hepatic mitochondrial adaptation (respiratory capacity, H_2_O_2_ production, mitophagy) in male and female mice fed a high-fat diet (HFD) and whether exercise (voluntary wheel running) affected outcomes [77]. They note wild-type female mice have an inherent ability to increase hepatic mitochondrial function in response to the obesogenic challenge, while males need exercise to equally adapt their respiratory capacity. In striking contrast to male mice, HFD-induced liver pathology is exacerbated by voluntary wheel running in females. Consistent with previous findings [52], reduced hepatic PGC-1α has a greater impact in female mice and led to higher liver triglycerides, markers of liver fibrosis and serum alanine aminotransferase (ALT) after high-fat diet feeding, yet reduction of the coactivator remarkably restored the benefits of exercise on the female liver. These data suggest that exercise has differential effects on hepatic metabolism in male and female livers, and while low hepatic PGC-1α worsens NAFLD in sedentary females, it may also improve their hepatic responses to exercise. Of note, PGC-1α-dependent effects on NAFLD in female mice within this study appeared independent of changes in mitochondrial respiration, implying that detrimental effects of reduced PGC-1α in the liver may be more influential on antioxidant capacity [52] or responses to inflammation [78,79,80]. PGC-1α expression is also negatively correlated with NAFLD severity in humans [81], but it remains to be seen whether this reflects a cause or a consequence of the mitochondrial abnormalities found in humans or whether targeting PGC-1α could be a viable option to treat the disease.

In addition to antioxidant enzymes, mitochondria also contain metabolic sensors such as the nicotinamide adenine dinucleotide (NAD^+^)-dependent histone deacetylase sirtuins that protect from the deleterious consequences of oxygen radicals on mtDNA, proteins, and lipids [82]. Sirtuin enzymes regulate biosynthesis, transport, and catabolism of NAD^+^, a dinucleotide cofactor with the potential to accept electrons in redox reactions [83]. NAD/NADH ratio is essential for mitochondrial ATP production and membrane potential and it appears that mitochondria have their own NAD biosynthetic machinery [84]. Three NAD-dependent deacetylase sirtuins (SIRTs) are localized in mitochondria: SIRT3, SIRT4, and SIRT5. SIRT4 is increased in NAFLD subjects, while the other sirtuins are decreased correlating with increased lipogenic genes including sterol regulatory element binding protein-1 (SREBP-1), fatty acid synthase, and acetyl-CoA carboxylase [85]. SIRT3 is the most investigated mitochondrial sirtuin and deficient SIRT3 activity predisposes mice to NASH [25,86]. SIRT3 is important for mitochondria biogenesis, carbohydrate metabolism, ketogenesis, amino acid metabolism, and stress-related pathways [87]. Deacetylation of the mitochondrial ROS scavenger superoxide dismutase 2 (SOD2) by SIRT3 in mice given streptozotocin (an ROS-inducer) points toward a protective role of this sirtuin in oxidative stress [88]. Moreover, reduced expression of SIRT3 and TFAM in diabetic hearts correlates with decreased transcription of mitochondrial DNA-encoded genes [89] and SIRT3 knockout mice have reduced fatty acid oxidation and low basal levels of ATP in the heart and liver [90,91]. Mice fed high-fat diets for a long time exhibit low SIRT3 activity, impaired mitochondrial function, and hyperacetylation of proteins in their livers [90]. Taken together, SIRT3 appears to play a protective role against NAFLD, possibly by improving mitochondrial function [92]. However, muscle- or liver-specific deletion of SIRT3 in mice subjected to a high-fat diet shows no difference in oxidative stress, glucose tolerance, or insulin sensitivity [93], and little is known about whether a gain-of-function in SIRT3 activity would have any beneficial effect.

### 2.3. Mitochondrial Networks, Housekeeping, and Cell Death

Chronic liver injury in individuals with NASH is associated with hepatocyte apoptosis [94,95]. Lipid peroxidation, formation of cytotoxic aldehyde, and proinflammatory cytokines activate stellate cells and hepatocyte cell death receptors (Fas, tumor necrosis factor receptor 1 (TNFR1), TNF-related apoptosis-inducing ligand (TRAIL)-receptor), resulting in fibrosis and apoptosis [96,97]. Activation of the mitochondrial intrinsic pathway of apoptosis (or programmed cell death) plays an important role in the progression of NASH [95]. Pro-apoptotic proteins of the B-cell lymphoma 2 (Bcl-2) family (e.g., bcl-2-associated X (Bax), bcl-2 homologous antagonist/killer protein (Bak), BH3 interacting-domain death agonist (Bid)) promote mitochondrial membrane permeabilization (MMP) by translocating to the mitochondrial outer membrane and forming mega channels. The anti-apoptotic proteins (e.g., Bcl-2, B-cell lymphoma-extra large (Bcl-xL), induced myeloid cell leukemia apoptosis regulator (Mcl-1)) prevent these processes [98]. Once triggered, MMP leads to the release of proteins such as apoptosis inducing factor (AIF) and cytochrome c which coordinate both mitochondrial dysfunction and trigger cell death pathways [99]. MMP can be influenced by lipid accumulation, ions (Ca^2+^), pH, ROS, and ATP levels [99]. In line with this, mice maintained on a high-fat diet and injected with glucose creating glucolipotoxic conditions displayed marked lipid peroxidation, hepatocyte apoptosis, and inflammation [100]. Moreover, mice maintained on a high-fat diet for eight weeks exhibited a disrupted mitochondrial respiratory chain and decreased ATP levels coupled with activated mitochondrial caspase 3-dependent cell death pathway [101]. Treatment with cyclosporin A, which inhibits the mitochondrial permeability transition via enhanced matrix Ca^2+^ buffering [102], can prevent mitochondrial dysfunction, oxidative stress, and hepatocyte apoptosis [100]. Taken together, there is a strong correlation between impaired mitochondrial function and cell death, and evidence suggests that their dysfunction amplifies the apoptotic signal in liver cells in response to high glucose and lipids.

Although there can be more mitochondria in a metabolically unhealthy liver, NASH is also characterized by the accumulation of abnormal mitochondria, which might be secondary to defects in mitophagy. Excess lipids, as well as insulin resistance and hyperinsulinemia, suppress autophagy by altering vesicular fusion, autophagy protein expression, and autophagy maturation [103]. Mitophagy is thought to protect against NAFLD, and its reduction in liver tissues from subjects with NASH correlates with disease severity [104]. Reduced mitophagy results in the accumulation of severely damaged mitochondria, leading to cell necrosis releasing mitochondrial damage-associated molecular patterns (DAMPs) that can promote liver inflammation and NASH development. Mitochondrial DAMPs (MTDs) activate pattern recognition receptors (PRRs), such as Toll-like receptor (TLR)9 [105,106]. After binding to TLR9, mitochondrial DAMPs trigger numerous downstream pathways, including nuclear factor kappa-light-chain-enhancer of activated B cells (NF-κB), the nucleotide binding oligomerization domain (NOD)-like receptor family, pyrin domain containing 3 (NLRP3), and interferon regulatory factor-dependent type 1 [107] to promote inflammation. In line with this, a decline in mitophagy is associated with activation of hepatic NLRP3 inflammasome in a murine mouse model of NASH and palmitic acid-treated primary hepatocytes [108], suggesting that decreased mitophagy might promote inflammation and NASH development.

While mitochondrial dysfunction is considered a critical component in NASH development, other potential mechanisms such as endoplasmic reticulum (ER) stress may contribute to disease progression. Both organelles physically interact through mitochondrial-associated membranes (MAMs) to dynamically adjust metabolic demand and responses to stress stimuli [109]. These contact sites normally allow the exchange of lipids, calcium (Ca^2+^), and ROS [110], and evidence suggests a relationship between MAM formation and the unfolded protein response (UPR), as ER stress sensors are found at the MAM interface and loss of MAM proteins induces UPR signaling [111,112]. ER stress markers are elevated in NASH-affected livers [113] and mitochondrial dysfunction correlates with excessive endoplasmic reticulum stress [30]. Mitochondrial ROS and oxidative stress can disrupt ER function leading to inappropriate release of calcium in liver parenchymal cells [114]. Recent studies also show that MAMs are important for hepatic insulin signaling, nutrient sensing, and glucose homeostasis [111,112]. Therefore, disruption of ER-mitochondria contact sites could exacerbate hepatic lipid accumulation and miscommunication between organelles appears to be related to the pathology of hepatic metabolic diseases. The complex interplay of cellular mechanisms impacting mitochondrial health and how this can lead to hepatocyte cell damage is illustrated in Figure 2.

### 2.4. We Are What We Eat

NAFLD is highly correlated with obesity and calorically high, nutritionally poor diets. Overconsumption of a diet rich in fat is also linked to the development of fatty liver disease [115]. Excessive storage of triacylglycerols in liver is a key feature of fatty liver disease and interestingly individuals with NASH have significantly more saturated fatty acids in their triacylglycerols [116]. Saturated fats promote fatty liver and accumulation of lipotoxic byproducts including ceramides and diacylglycerols [117,118] is associated with hepatic inflammation and mitochondrial ROS production resulting in liver cell death [119]. Moreover, in vitro treatment of liver cells with saturated fatty acids (palmitic and stearic acids) reproduces mitochondrial dysfunction found in NASH, including decreasing cellular ATP content and mtDNA-encoded oxidative phosphorylation (OXPHOS) subunit expression, coupled with increased oxidative stress [120].

In addition to fat, excessive glucose consumption (or hyperglycemia associated with conditions such as diabetes) could be a primary cause of mitochondria respiratory chain dysfunction by increasing the burden on oxidative phosphorylation and causing oxidative stress [121]. Vanhorebeek et al. reported that normoglycemia or strict control of blood glucose in humans prevents or reverses ultrastructural or functional abnormalities of hepatocyte mitochondria [121]. Consistent with this, high glucose induces mitochondrial fragmentation, ROS production, loss of mitochondrial membrane potential, and ATP depletion in several rodent and human cell models [122,123,124,125,126,127]. In addition to glucose, fructose appears to be particularly harmful to liver mitochondria [128]. In addition to its ability to promote de novo lipogenesis and block β-oxidation of fatty acids, fructose consumption seems to cause a drop in ATP and an elevation of uric acid, which can further induce mitochondrial oxidative stress [129]. Moreover, a diet rich in fructose is associated with increased oxidative mtDNA lesions in rat liver coupled with reduced mitochondrial repair capacity [130] and rats consuming a diet high in fat and rich in fructose have increased hepatocyte damage, inflammation, and lipid peroxidation coupled with impaired mitochondrial respiration and activity [131]. Studies show that dietary fructose induces fatty liver [132,133,134] and inflammation in mice after 8–24 weeks of exposure [52,135,136], as well as hepatic fibrosis in rhesus monkeys after seven years [137]. While these data in animals and humans implicate fructose as a risk factor for fatty liver [138], further studies are needed to investigate the direct relationship between mitochondrial dysfunction and fructose consumption in humans.

Dietary cholesterol is also implicated in the development of NASH in mouse models and humans [139]. More precisely, dietary-induced hypercholesterolemia causes oxidative stress, loss of mitochondrial membrane potential, reduction in ATP content, loss of mitochondrial cristae, and hepatic steatosis in mice [140]. It is also suggested that overload of free cholesterol disrupts mitochondrial and ER membrane integrity, triggering mitochondrial oxidative injury and ER stress [141]. Bellanti et al. showed through targeted lipidomic analysis that rats fed diets rich in cholesterol exhibited increased toxic hepatic triol, which induces apoptosis and impaired mitochondrial respiration in vitro [142]. Gan et al. showed that mitochondrial-free cholesterol deposition causes hepatocyte apoptosis and necrosis through c-Jun N-terminal kinase (JNK) activation, associated with increased high-mobility group box 1 (HMGB1), and cytolytic effects on neighboring hepatocytes driven by Toll-like receptor 4 (TLR4) [143]. Moreover, mitochondrial-free cholesterol sensitizes liver cells to TNFα- and Fas-mediated steatohepatitis and causes mitochondrial-reduced glutathione (mGSH) depletion [144], which is also reported in animal models and patients with NASH [145]. Thus, accumulating evidences suggest that cholesterol overload in mitochondria induces redox imbalances leading to oxidative stress and cell death associated with steatohepatitis [146,147,148]. Consumption of high calorie diets rich in sugar, saturated fat, and cholesterol contributes to obesity, a major risk factor of fatty liver disease. While disease in the liver may simply be due to accumulating lipids, evidence suggests that the direct effect of these nutrients and their metabolism on mitochondrial health, independent of weight gain, likely plays a significant role in pathogenesis.

## 3. Mitochondrial Dysfunction in NASH-Associated HCC

### 3.1. From Cell Death to Proliferation

Prolonged exposure of hepatocytes to lipids and inflammation can result in the exhaustion of liver defense mechanisms, leading to chronic liver disease that promotes the development of hepatocellular carcinoma (HCC). Multiple mechanisms are proposed for the pathogenesis of obesity-associated HCC [149], and they can be different from cancers stemming from alcohol or viral origins. Among these, mitochondrial dysfunction is linked to cancer progression through increased ROS production, impaired mitochondrial respiration, ER stress, and alteration of nutrient metabolism [150]. Overproduction of ROS impairs the mitochondrial respiratory chain, leading to cytochrome c release, and apoptotic death signals in cancer cells as well [151]. Moreover, oxidative stress and ROS lead to the release of calcium by the endoplasmic reticulum, further increasing mitochondria ROS production and potentiating proapoptotic pathways [151,152]. While apoptosis impedes the growth of hepatocellular carcinoma cells [153], it is also believed to promote cancer initiation through compensatory proliferation of progenitor cells in the liver [154,155]. This increased proliferation within an environment of higher ROS production and DNA damage causes gene mutations to accumulate in mature and newly formed hepatocytes. As time passes, cells that can now resist apoptotic pressure and escape cell cycle checkpoints persist and contribute to malignant transformation and cancer development [50]. Furthermore, an environment where mitochondrial function is already reduced by metabolic stress could favor survival of cells that do not depend on mitochondria for energy production, a feature of many cancer cell types [156]. Lastly, damage or mutation of mtDNA itself may potentiate HCC. A low mtDNA copy number is associated with liver cancer and this may confer resistance to chemotherapy [157,158]. Thus, whether it be ROS-induced accumulation of DNA mutations (both in mtDNA and ncDNA), increased mitochondrial-mediated apoptosis due to metabolic exhaustion, or a combination of the two, mitochondrial abnormalities can contribute in many ways to cancer development in obesity and metabolic disease.

Just like in NASH, metabolic stress, mitochondrial dysfunction, and oxidized mtDNA also indirectly contribute to the progression of HCC through influences on inflammation and immunologic pathways, including activation of the inhibitor of nuclear factor kappa-B kinase subunit beta (IKKβ)/NF-κB signaling pathway, a key node that plays a crucial role in hepatocyte survival and inflammatory responses [154,158]. ROS and DNA/lipid peroxidation increase the release of cytokines including TNF-α and IL-6, which activate pro-oncogenic pathways via JNK, signal transducer and activator of transcription 3 (STAT3), Janus kinase 2 (JAK2), MAPK, and phosphoinositide 3-kinase (PI3K) [159,160,161]. TNF-α and IL-6 promote the development of obesity-induced HCC development in mice exposed to the liver carcinogen diethylnitrosamine when combined with a high-fat diet [159]. Moreover, TNF-α and IL-6 promote iron accumulation, which further fuels oxidative stress-driven cell toxicity and can activate fibrogenesis and carcinogenesis in metabolic syndrome [162]. Indeed, detrimental effects of hepatic iron overload seem to impact HCC development in NASH patients [163,164]. Thus, mitochondrial dysfunction, in combination with or driving cytokine imbalances and iron accumulation, are part of a damaging cycle that potentiate hepatocyte injury, NAFLD progression, and malignant transformation of hepatocytes.

The production of ROS also stabilizes hypoxia inducible factor alpha (HIFα) subunits by inactivating prolyl hydroxylase domain-containing protein (PHD) enzymatic activity [165]. Since α-ketoglutarate is needed for PHD function, conditions promoting low concentrations of this metabolite (another consequence of mitochondrial dysfunction) will lead to an overall increase in HIF transcriptional activity [165]. HIFα subunits are stable under hypoxia, while they are rapidly degraded under normal oxygen tension. Changes in oxygen tension occur during steatohepatitis, and when combined with inflammation, may be sufficient to promote a hypoxic response that increases HIF-regulated transcription of genes involved in cellular metabolism, angiogenesis, and proliferation, all of which promote tumor development [165]. Moreover, a hypoxic environment further compromises ATP generation, leading to heavier reliance on glycolysis [166]. In agreement with this, mice with hepatocyte-specific deletion of HIF-2α have reduced liver tumor numbers and size on a choline-deficient L-amino acid refined diet following diethylnitrosamine administration [167]. Additionally, increases in hepatic HIF-1α accelerate the transition from NASH to HCC in a model of NASH-associated liver cancer involving a high-fat–high-cholesterol–high-sugar diet combined with diethylnitrosamine [168]. Thus, mitochondrial dysfunction could promote the transition from NASH toward HCC by potentiating pathways designed to protect cells from hypoxic damage that unfortunately also promote tumor development within permissive environments.

### 3.2. Cancer Cells: It is Not Only What You Burn, But When

Metabolic reprogramming is a key event in HCC development and progression [28,29]. Metabolites generated during this process can be cancer-promoting and favorable metabolite signatures seem to be selected for tumor cells [28]. Well-characterized metabolic changes thought to promote cancer include the Warburg effect (a switch toward anaerobic glycolysis) [169], increased glutamine metabolism, and defects in one-carbon metabolism [170]. Early studies in malignant tumors describe a fundamental reprogramming of gene expression resulting in a highly glycolytic “Warburg” phenotype and suppression of mitochondrial biogenesis. However, new evidence suggests that because of sustained proliferation and nutrient depletion in malignancies, a second “wave” of cancer cell adaptation is required to restore the more efficient process of oxidative phosphorylation to accommodate increasing energy demands [171]. Interestingly, increased lactate production is observed in NASH patients, suggesting there may be a shift toward a glycolytic phenotype in the first steps of liver carcinogenesis [172]. In the second steps of malignant transformation, increased cellular dependency on glutamine can occur to support the high bioenergetic demand, generating high levels of α-ketoglutarate and citrate to support the TCA cycle to provide carbon cycles for amino acid biosynthesis [173].

Proliferative tissues are highly sensitive to changes in one-carbon (folate) metabolism and dysregulation of this pathway is reported in NASH and HCC [174,175]. Mitochondrial one-carbon metabolism is required for several biological reactions including nucleotide synthesis, amino acid metabolism, NADPH production, and epigenetic methylation [176]. Alterations in major sources of one-carbon units such as glutathione, choline, carnitine, serine, and glycine compromise mitochondrial performance and metabolic control [174]. For instance, serine is the most important donor of one-carbon units through the mitochondrial serine hydroxymethyltransferase (SHMT2), which catalyzes the reversible conversion of serine to glycine [174]. Although the conversion rate of serine has not been reported in obesity and associated metabolic disorders, decreased plasma level of serine is seen in NASH patients [177,178,179,180]. In contrast, serum levels of serine are increased in humans with HCC compared with healthy subjects [181,182] and high SHMT2 expression is associated with negative prognosis in human hepatocellular carcinoma [183]. Downregulation of SHMT2 suppresses human HCC cell proliferation and liver tumor incidence and growth in a xenograft model [184]. It is also suggested that cancer cells benefit from enhanced serine-dependent one-carbon metabolism for the production of NADPH [185] and this may be exacerbated by obesity. Disturbances in liver one-carbon metabolism, including increases in transcript level of mitochondrial SHMT2, are associated with hepatic lipid accumulation in mice fed a high-fat diet [186].

Altered or enhanced lipid metabolism is also implicated in metabolic reprogramming of cancer cells [187]. A lipid-rich condition characterized by large droplet steatosis and ballooning, and associated with pericellular fibrosis and inflammation, is observed in tumor cells of steatohepatitic variants of HCC [188]. Metabolic profiling by liquid chromatography-mass spectrometry (LC-MS) reveals extensive accumulation of long-chain acylcarnitine species in HFD-fed HCC tissues of mice correlated with the increased expression of the mitochondrial transporter carnitine palmitoyltransferase-1 (CPT1) and decreased expression of CPT2, an important enzyme of the mitochondrial long-chain fatty acid oxidation [28]. Lipid metabolic reprogramming mediated by CPT2 downregulation in HCC cells can promote liver cancer through accumulation of acylcarnitine as an oncometabolite [28]. Indeed, decreases in CPT2 are reported in HCC and in subjects with chemoresistance to cisplatin [189]. These findings are in line with previous studies showing that the expression of acylcarnitine metabolism-related genes are altered in NASH-driven HCC mouse models, including HFD-fed major urinary protein (MUP)- urokinase-type plasminogen activator (uPA) [190] mice and phosphatidylinositol 3-kinase catalytic subunit alpha PI3KCA transgenic mice [191]. Overall, mitochondria are involved in several biochemical pathways that generate metabolic precursors needed to support the sustained bioenergetic demands of NASH and HCC pathogenesis.

### 3.3. Transcriptional Regulation of Mitochondrial Adaptation in HCC

Given the role of PGC-1α as a key regulator of mitochondrial metabolism, adaptation, and antioxidant defense, several studies have investigated its role in cancer development, including hepatocellular carcinoma. Whether it sustains or interferes with tumorigenesis remains contradictory. Reduced hepatic PGC-1α expression is reported in human HCC associated with increased glycolysis and reduction of blood glucose [192]. Decreased PGC-1α is also associated with dedifferentiation of human hepatoma cell lines through impairment of hepatocyte nuclear factor 4 alpha (HNF4α), a transcription factor critical for liver development [193]. Consistent with the coactivator having tumor-suppressive functions, Lee et al. showed that overexpression of PGC-1α increases the epithelial marker E-cadherin and reduces motility of HepG2 cells possibly through PPARγ [194] and PGC-1α can bind p53 and enhance transcription of proarrest genes including cyclin-dependent kinase inhibitor 1 (p21), growth arrest and DNA damage-inducible 45 (GADD45), TP53-inducible glycolysis and apoptosis regulator (TIGAR), SCO cytochrome C oxidase assembly Protein 2 (SCO2), and sestrin2 in human hepatoma cells [195]. On the other hand, Ballah et al. showed that loss of PGC-1α in whole-body knockout mice prevented diethylnitrosamine (DEN)-induced liver cancer development and overexpression of the coactivator promotes tumor development in a xenograft model by increasing lipogenesis [196]. Interestingly, mice with liver-specific overexpression of the related coactivator PGC-1β also exhibited increased chemically-induced (DEN) and genetically-induced (Abcb4^−/−^) liver cancer resulting from increased ROS scavenging and tumor anabolism [197]. Interestingly, the authors’ report reduced PGC-1α in liver tumors of these mice and, vice versa, mice with liver-specific ablation of PGC-1α had a compensatory increase in PGC-1β [52]. Although PGC-1α and PGC-1β are often viewed as redundant in function, there is evidence in liver that they may regulate different gene programs related to nutrient metabolism, namely PGC-1β is the primary isoform-enhancing lipogenesis [52,81,197,198,199]. This may suggest that PGC-1β is a driver of tumor formation, while PGC-1α confers resistance. However, although there is strong evidence linking PGC-1 coactivators to HCC pathogenesis, it still remains unclear how this family of coactivators directly affects cancer cell biology and tumor development, particularly when metabolism is altered as in obesity or metabolic disease.

Answering this question is complicated by the multiple ways by which PGC-1 activity is controlled in cells. In addition to changes in mRNA and protein level, post-translational modification of PGC-1α has been implicated in hepatocarcinogenesis, including its phosphorylation by AMPK and deacetylation by sirtuins [29]. Zhang et al. showed that activation of the AMPK-PGC-1α pathway inhibits proliferation and induces apoptosis of hepatocellular carcinoma cell lines [200]. Li et al. showed that SIRT1/PGC-1α axis facilitates hepatocellular carcinoma metastasis through increased mitochondrial biogenesis [201]. Apart from the deacetylase activity of SIRT1 on PGC-1α, less is known about the impact of PGC-1α on sirtuins themselves. Kong et al. showed that SIRT3 is a downstream target gene of PGC-1α in hepatocytes and mediates the suppressive effects of PGC-1α on ROS production [202]. SIRT3 can also inhibit growth and proliferation of the human hepatoma cells HepG2 and induce apoptosis [203] and is a tumor suppressor in the human hepatoma cells Huh7, reducing phosphorylation of PI3K/Akt [204]. Of note, SIRT3 expression is downregulated in human HCC [203,204] and low SIRT3 expression is associated with poor differentiation and unfavorable prognosis [205]. Thus, mitochondrial SIRT3 appears to play a protective role in HCC. Given the potential link between PGC-1α and SIRT3, it could be interesting to investigate the role of this partnership in NAFLD-associated liver cancer. Taken together, studies suggest that PGC-1α may have different roles in tumor development depending on the cell type, environment, and current metabolic state. Further studies are needed to clarify whether PGC-1α plays a significant role in hepatic cancer development and whether this role changes in different metabolic environments based on the need or dependence of cancer cells on mitochondrial metabolism.

## 4. Challenges in Targeting Mitochondria for NASH-Related HCC Treatment

Currently, the tyrosine kinase inhibitor sorafenib is the only approved systemic medication for the treatment of HCC, increasing patient survival by months [206]. Chemotherapy and radiotherapy are generally ineffective, as HCC cells are highly chemoresistant. There is critical need for new effective and more targeted therapies for this disease. As evidence in animal models supports mitochondrial dysfunction as a key player in metabolic liver disease pathogenesis, targeting mitochondria represents an attractive strategy to stop or slow down NASH-related HCC progression. However, what has been revealed from these efforts is at times paradoxical. Mitochondrial-based therapies do indeed have varying efficacy to treat liver disease and cancer, though is not clear whether the correct strategy should be to boost or inhibit their function. There is currently no available treatment specifically targeting mitochondria in NASH-related HCC. We will discuss this as a potential treatment strategy, its associated challenges, and paradoxical findings, and avenues that remain to be investigated.

### 4.1. Lifestyle Intervention

Regular exercise and caloric restriction are the only physiological interventions proven to reduce steatosis, inflammation, and fibrosis in NAFLD patients [207]. Mechanistic studies in mice show that exercise improves liver mitochondrial morphology, and increases mitochondria biogenesis (PGC-1α and TFAM), autophagy-related proteins, and SIRT3 [208,209]. However, whether these outcomes on mitochondrial health help to prevent HCC in humans is extremely difficult to determine. A large prospective study suggests that increased physical activity reduces liver cancer risk in men; however, the etiology of liver cancer in this study was not restricted to NASH [210]. On the other hand, rapid weight loss with severe dieting or malnutrition increases liver inflammation and fibrosis [211]. Unfortunately, despite the known benefits of caloric restriction to reduce obesity and lengthen lifespan, lifestyle modification is very difficult to implement, and even more difficult to maintain, in the general population. Notably, since the diagnosis of liver cancer is often made late in its progression when overall health and energy of the patient are already failing, diet and exercise may not always represent a recommendable or feasible course of action.

### 4.2. Antioxidants

Since both increased mitochondrial function and dysfunction causes ROS production, antioxidant supplementation is predicted to prevent steatohepatitis and possibly have anti-cancer effects. However, broad specificity of this therapy may also have detrimental effects on cellular pathways that require ROS, such as redox signaling [212], a caveat not often considered during evaluation of these approaches. Indeed, the reduced glutathione (GSH) precursor N-acetylcysteine (NAC) prevents development of steatosis in methionine-choline deficient (MCD)-diet fed rats and ob/ob mice [213,214]. It also attenuates hepatocarcinogenesis in TLR2-deficient mice by inhibiting ROS and ER stress, and induces apoptosis in human liver cancer cells [215,216]. Vitamin E (α-tocopherol) can act as a peroxyl radical scavenger [217,218]. Deficiencies in vitamin E can be found in NASH patients [219] and the PIVENS (Pioglitazone, Vitamin E or Placebo for Nonalcoholic Steatohepatitis) trial in NASH patients showed a reduction in hepatic steatosis, lobular inflammation, and transaminases [220]. Interestingly, vitamin E treatment in children with NAFLD may provide no improvement of liver function [221]. Based on these results, the American Association for the Study of Liver Disease (AASLD) recommended in 2012 that vitamin E should be considered for non-diabetic adults with biopsy-proven NASH [222]. Vitamin E prevents HCC in mice via downregulation of inducible nitric oxide synthase (iNOS) and nicotinamide adenine dinucleotide phosphate (NADPH) oxidase [223] and protects against oxidative DNA damage in human HCC cell lines [224]. Zhang et al. also reported that dietary vitamin E decreases the risk of developing hepatitis B virus (HBV)-related HCC in humans [225]. Finally, while antioxidants may help prevent cancer initiation by reducing ROS, they may also lessen ROS to levels that support cancer cell proliferation and metastasis by minimizing some of the negative effects (i.e., DNA damage) of ROS on malignant cells, thus cancer stage and whether ROS is a major driver influence outcomes. Interestingly, a very effective method to reduce cellular ROS is to inhibit mitochondrial oxidative phosphorylation with mitochondrial toxins (e.g., metformin) or mild chemical uncouplers. Inhibition of mitochondrial function with these chemicals also effectively reverses NAFLD and prevents HCC in mice [226] and zebrafish [227], arguing that it might not be reduced mitochondrial function or oxidative capacity that is pathogenic per se, but rather an inability to effectively deal with the bioproducts of their activity.

### 4.3. Pioglitazone

The antihyperglycemic drug pioglitazone is an interesting candidate for the treatment of NASH [228]. It belongs to the thiazolidinedione family and acts as a potent ligand for the nuclear factor peroxisome proliferator-activated receptor gamma (PPARy). Cusi et al. showed that 18 and 36 months of pioglitazone treatment in prediabetic or type 2 diabetic patients with biopsy-proven NASH improved fibrosis score, hepatic triglyceride content, as well as hepatic and muscle insulin sensitivity [229]. It also ameliorates steatosis and necroinflammation in humans and diabetic mice with NASH [230,231] and improves hepatic mitochondrial oxidative capacity in a mouse model of NASH [232]. It effectively delays liver fibrosis and hepatocarcinogenesis in two rodent models of HCC induced by diethylnitrosamine alone or diethylnitrosamine combined with a choline deficient, L-amino acid defined, high-fat diet [233]. Interestingly, thiazolidinediones including pioglitazone lower the risk of liver cancer incidence and HCC recurrence in diabetic patients [234,235]. Whether the beneficial effects of this drug are due to improved hepatic mitochondria function, or the many other known benefits of PPARγ activation such as improved insulin sensitivity, remains to be determined. However, concerns over possible hepatotoxicity and other undesirable side effects make PPARγ agonists unlikely to be used widely for NASH or HCC, or at least not in their current form.

### 4.4. NAD^+^ Precursors

Boosting mitochondrial redox homeostasis using NAD^+^ precursors may also be attractive. For example, resveratrol increases the NAD^+^/NADH ratio by activating AMPK in combination with SIRT1 or SIRT3, resulting in increased fatty acid β-oxidation and tricarboxylic acid cycle in liver cells [236]. Resveratrol also reduces oxidative stress, inflammation, fibrosis, hepatic lipogenesis, and steatosis in animal models of NASH and NAFLD [237], and in obese persons [238]. On the other hand, resveratrol impairs F0/F1 ATP synthase activity in mitochondria isolated from rat brain and liver, which is suspected to result in cell death by apoptosis or necrosis [239]. While the pro-apoptotic effect of resveratrol could be useful for cancer cell eradication [240], this could also be harmful for healthy cells. In humans, its effectiveness as a treatment for NAFLD remains inconclusive, as resveratrol has only been tested in a small number of clinical trials [241]. Instead, it is suggested that it might be more effective as supplementation to additional lifestyle changes. Finally, it is important to note that the bioavailability of resveratrol is very limited in humans. Other approaches to boost NAD^+^ levels include inhibition of NAD^+^-consuming enzymes (such as poly ADP-ribose polymerase; PARPs and CD38) and supplementation with NAD^+^ precursors [242,243]. Inhibition of the glycohydrolase CD38 in mice protects from HFD-induced obesity and liver steatosis via NAD-dependent activation of the SIRT1-PGC-1α pathway [244]. Importantly, PARP1 is robustly activated in the liver of NAFLD patients and positively correlates with hepatic steatosis (i.e., triglycerides and free fatty acids) [245]. Mechanistic studies in mice show that inhibition of PARP protects from NAFLD and NASH [246,247]. Gariani et al. demonstrated that it replenishes NAD^+^ levels and promotes mitochondrial biogenesis and hepatic β-oxidation [246]. Supplementation with the NAD precursor pyridine nucleoside form of vitamin B3, nicotinamide riboside, protects and reverts NAFLD through SIRT1- and SIRT3-dependent mitochondrial unfolded protein response, increasing TCA cycle, OXPHOS, and hepatic β-oxidation [248]. Nicotinamide riboside supplementation also reduces body weight and attenuates liver fibrosis in a diet-induced mouse model of NAFLD [249]. Since NAD^+^ boosters have shown efficacy in rodents and appear to raise NAD^+^ levels in mouse and humans [248,250,251,252,253], research interest into the effects of NAD^+^ precursors in humans is growing considerably. However, efficacy in subjects with metabolic disorders remains to be determined. Boosting NAD^+^ for the prevention and treatment of liver cancer has also been proposed [254,255]. However, the effects of nicotinamide ribosides on mitochondria in the context of HCC remains to be investigated.

### 4.5. Master Regulators of Metabolism

Another potential therapeutic strategy could be to use AMPK activators such as the 5-aminoimidazole-4-carboxamide ribonucleotide (AICAR), which can increase insulin sensitivity, improve glucose uptake, and reverse hepatocyte injury [58,256]. AICAR inhibits hepatosteatosis and liver tumor development in a mouse model of HFD-related HCC [257] and AICAR reduces human HCC cell adhesion and proliferation via cell cycle arrest in G1/S [258,259]. However, similar to resveratrol, AICAR is poorly bioavailable when administered orally [260] and can also induce mitochondrial-mediated apoptosis [261,262]. Given the diverse and numerous effects of AMPK activation on cells in general, in addition to their low bioavailability, the use of these molecules requires a more in-depth understanding of the downstream molecular mechanisms implicated in their anti-NAFLD effects and whether these would be different in cancer cells versus differentiated liver cells. PGC-1α may also represent an interesting target, as it regulates a vast transcriptional network of pathways centered on enhancement and maintenance of mitochondrial biogenesis, oxidative capacity, fatty acid oxidation, and ROS detoxification. However, targeting PGC-1α is not easy, as in contrast to nuclear receptors, enzymes, or cell-surface receptors, transcriptional coactivators lack highly specific ligand-binding domains [81]. Interestingly, many treatments acting through SIRT1 and AMPK, such as the AICAR, metformin, and resveratrol have been shown to induce or activate PGC-1α in different organs [263,264,265,266,267]. However, PGC-1α-dependent effects of these molecules remain to be determined.

An important final consideration for all NASH and HCC therapies is whether the duration of treatment reverses and/or maintains improved liver health. Clinical studies reporting improvements in NAFLD and prevention of HCC are generally performed over short periods of time (i.e., over two years) [268]. Given that the progress to HCC from NAFLD/NASH occurs over a relatively long period (10–20 years), long-term safety and efficacy for any treatment for NASH and associated liver cancer would need to be established. In addition, lifestyle changes including weight loss, exercise, and diet modification are likely needed to prevent reoccurrence of the disease. More research is needed to understand how these environmental and lifestyle modifications affect treatment efficacy and safety, as level of compliance (ranging anywhere from intense intervention to minimal participation) is hard to control in humans.

## 5. Conclusions

Mitochondria play a key role in liver metabolism and likely lie at the core of NASH. They are important for adaptation to nutrient overload through their ability to increase β-oxidation of fatty acids and oxidative phosphorylation. However, this comes at a considerable cost, as increases in activity increases ROS production that can lead to apoptosis and compensatory cell proliferation contributing to malignant transformation. Emerging evidence clearly suggests that mitochondrial dysfunction is central to NASH and HCC. Recent advances in animal models shed some light on the complex network involved in the pathogenesis with many pointing toward the significance of inappropriate ROS regulation, but replicating the full spectrum of human NASH-HCC in rodents remains challenging, as we still do not yet know the precise diet, lifestyle, and environmental variables that trigger NASH in humans, let alone other species. Determining whether the mitochondrial defects observed in the NASH-HCC transition translate into causative explanations for the disease is a major priority, as it is at this stage of the disease where targeting will have the most benefit. It is difficult to dismiss the abundance of evidence linking mitochondrial bioenergetics, ROS imbalance, ER stress, inflammation, and cell death to metabolic disease and carcinogenesis. Thus, the therapeutic potential of agents targeting mitochondrial function in NASH-HCC transition remains a very interesting avenue to pursue, but it appears that there is still much more we can learn about these important organelles, particularly within the setting of human disease.

## Figures and Tables

**Figure 1 metabolites-09-00233-f001:**
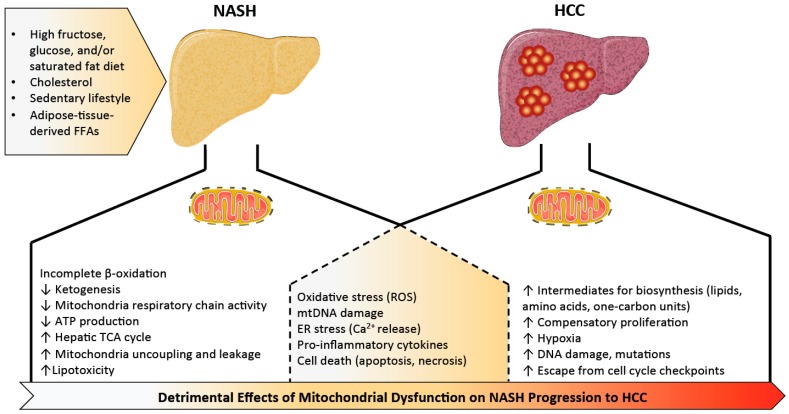
Roles for mitochondria in the progression of steatohepatitis to hepatocellular carcinoma. Metabolic stress induced by calorically high, nutritionally poor diets leads to metabolic disturbances in hepatic mitochondria. Mitochondrial adaptation and flexibility become compromised once (or possibly just before) the disease progresses to steatohepatitis (NASH). This results in incomplete β-oxidation, impaired ketogenesis, reduced mitochondria respiratory chain activity and ATP production, coupled with overactive TCA cycle potentially to meet the high energy demand. Sustained mitochondrial oxidative flux results in increased ROS production associated with mtDNA damage, ER stress, tissue inflammation and eventual cell death. Just like in NASH, oxidative stress, mitochondrial dysfunction, and oxidized mtDNA also contribute to the progression of HCC through influences on inflammation, cell death, and ER stress. As time passes, changes in lipid metabolism, one-carbon metabolism, and amino acid biosynthesis take place, fueling cancer cells for compensatory proliferation and further enhancing hypoxia, DNA damage, mutations, and escape from cell cycle checkpoints. NASH: nonalcoholic steatohepatitis, ER: endoplasmic reticulum, TCA: tricarboxylic acid cycle, HCC: hepatocellular carcinoma, mtDNA: mitochondrial DNA, ATP: adenosine triphosphate, ROS: reactive oxygen species, FFA: free fatty acids.

**Figure 2 metabolites-09-00233-f002:**
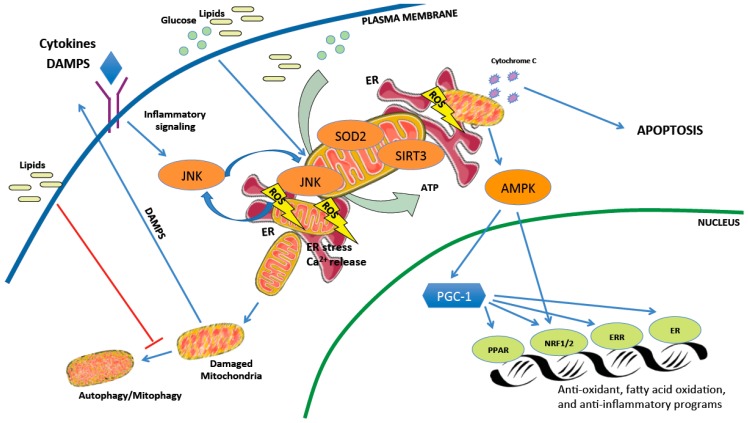
Mitochondrial pathways implicated in the pathogenesis of NASH and HCC. Mitochondria are responsible for converting nutrients such as lipids and glucose into energy (ATP). Mitochondrial activity leads to an increase in reactive oxygen species (ROS) production, which are detoxified by anti-oxidant defenses including SIRT3 and SOD2. Decreased mitochondrial efficiency (accumulation of ADP) and ROS activate AMPK and transcriptional machinery, including PGC-1α, to stimulate expression of gene pathways promoting mitochondrial adaptation. Damaged mitochondria are usually eliminated by autophagy/mitophagy which is attenuated by lipid accumulation. Prevention of autophagy further leads to accumulation of damaged mitochondria that can release inflammatory DAMPs and cytochrome C to promote cell death, and/or exacerbate endoplasmic reticulum (ER) stress. SIRT3: NAD+-dependent deacetylase sirtuin-3, SOD2: superoxide dismutase 2, ADP: adenosine diphosphate, AMPK: AMP-activated protein kinase, PPAR: peroxisome proliferator-activated receptor, NRF1/2: nuclear respiratory factor 1/2, ERR: estrogen related receptor, JNK: c-Jun N-terminal kinase, PGC-1: peroxisome proliferator-activated gamma coactivator-1, DAMPs: damage-associated mitochondrial molecular patterns.

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
