# Peer review of "Mitochondrial Dysfunction in the Transition from NASH to HCC"

_metabolites, 2019, doi:10.3390/metabo9100233_

Round 1

Reviewer 1 Report

In this review the authors elaborate several potential mechanisms that could lead to simple steatosis, nonalcoholic steatohepatitis and hepatocellular carcinoma, along with some of the beneficial therapeutic strategies. Below are some of the major concerns from this reviewer.

The authors have tried to discuss three major variations or stages of chronic liver disease, namely simple steatosis, nonalcoholic steatohepatitis and hepatocellular carcinoma all under the same discussion. While there are potentially several mechanisms contributing to the origin of these disease states, their discussion under a common umbrella, makes it unclear and very confusing for the reader. The authors should identify a priority for either nonalcoholic steatohepatitis or hepatocellular carcinoma, and restrict their major discussion to one of these. The review lacks any kind of schematic or figures or tables to lead the readers to a focused idea or conclusion as they go through the discussion. This aspect is particularly relevant as the authors present a lot of discrete and diverse mechanisms which could contribute to these chronic liver diseases. Significant aspects of mitochondrial function which has received major attention in the field as contributors to the etiology of mitochondrial dysfunction during NAFLD has not even been mentioned in the discussion. For example, there are several high impact human and animal studies illustrating a clear role for the mitochondrial oxidative networks (e.g. Kreb’s cycle activity, Ketogenesis, Oxphos etc) and lipogenesis towards the etiology of these chronic diseases. Furthermore, the interactions of these biochemical networks with processes of inflammation, lipotoxicity and other lipid storage/ disposal mechanisms are also not discussed. These aspects are emerging therapeutic targets and thus need to be included in the discussions. The review spends a large portion of time discussing about mechanisms of inflammation, ROS production etc. However considering the title and the objective of the review, the mechanisms linking mitochondrial dysfunction and these processes are not explicitly stated.   This reviewer believes that pooling together the treatment strategies for nonalcoholic steatohepatitis and hepatocellular carcinoma, without a strong argument for one versus the other, is misleading. Even though some of the treatment strategies have beneficial effects towards both nonalcoholic steatohepatitis and hepatocellular carcinoma, the authors should carefully emphasize the relative significance (based on available evidence) of each of these treatments towards improving nonalcoholic steatohepatitis and hepatocellular carcinoma.

Author Response

We appreciate the time and effort taken by the reviewer and have incorporated the suggested changes. Most notable, we have added two figures that illustrate the main ideas and focus of the manuscript. Our responses to individual comments are addressed below in blue.

1) The authors have tried to discuss three major variations or stages of chronic liver disease, namely simple steatosis, nonalcoholic steatohepatitis and hepatocellular carcinoma all under the same discussion. While there are potentially several mechanisms contributing to the origin of these disease states, their discussion under a common umbrella, makes it unclear and very confusing for the reader. The authors should identify a priority for either nonalcoholic steatohepatitis or hepatocellular carcinoma, and restrict their major discussion to one of these.

We have clarified our focus to the role of mitochondria in the transition from NASH to HCC, and modified our title accordingly. We are particularly interested in the role of mitochondrial dysfunction in this important pathological transition and have tried to reorganize the manuscript to involved discussion of mitochondrial-related pathways shown to be important for this aspect of disease.

2) The review lacks any kind of schematic or figures or tables to lead the readers to a focused idea or conclusion as they go through the discussion. This aspect is particularly relevant as the authors present a lot of discrete and diverse mechanisms which could contribute to these chronic liver diseases.

We have now added 2 figures to our review to help illustrate our main points. Figure 1 shows the role of mitochondrial in each disease (NASH and HCC) and the overlapping pathways that have been implicated in the transition. Figure 2 illustrates the major cellular mechanism related to mitochondrial dysfunction implicated in this transition.

3) Significant aspects of mitochondrial function which has received major attention in the field as contributors to the etiology of mitochondrial dysfunction during NAFLD has not even been mentioned in the discussion. For example, there are several high impact human and animal studies illustrating a clear role for the mitochondrial oxidative networks (e.g. Kreb’s cycle activity, Ketogenesis, Oxphos etc) and lipogenesis towards the etiology of these chronic diseases. Furthermore, the interactions of these biochemical networks with processes of inflammation, lipotoxicity and other lipid storage/ disposal mechanisms are also not discussed. These aspects are emerging therapeutic targets and thus need to be included in the discussions. The review spends a large portion of time discussing about mechanisms of inflammation, ROS production etc. However considering the title and the objective of the review, the mechanisms linking mitochondrial dysfunction and these processes are not explicitly stated.  

We agree that we had failed to cover important aspects of mitochondrial dysfunction in metabolic liver disease. Others have nicely reviewed this literature previously for NAFLD and NASH pathology and we have referred to this in our revised version. Our goal was to focus primarily on the specific aspects of mitochondrial dysfunction important for the NASH to HCC transition, which we have now clarified in the text and using Figure 1.

4) This reviewer believes that pooling together the treatment strategies for nonalcoholic steatohepatitis and hepatocellular carcinoma, without a strong argument for one versus the other, is misleading. Even though some of the treatment strategies have beneficial effects towards both nonalcoholic steatohepatitis and hepatocellular carcinoma, the authors should carefully emphasize the relative significance (based on available evidence) of each of these treatments towards improving nonalcoholic steatohepatitis and hepatocellular carcinoma.

Our goal in this section was to discuss potential strategies targeting mitochondrial health in NASH to prevent transition to HCC. There is some clinical evidence to support this theory in both NASH and HCC, which was our justification for discussing the two diseases. We have modified the text to clarify our rationale for this in the introductory paragraphs to the section.

Please see the attached marked-up version of the revised manuscript, where you can review the changes made. We thank the reviewer for their helpful and constructive comments, which we feel have significantly improved the manuscript. We hope that these revisions are sufficient to warrant publishing of the review article.

Reviewer 2 Report

In this review, Leveille and Estall summarize functional roles of mitochondria in carcinogenesis of HCC. This review covers well for mitochondrial dysfunction in NASH and I do not have any comments for this manuscript. However, it would be useful for readers if the authors provide 1-2 figures to represent pathways/associations because there are so many genes and pathways are shown in this manuscript, which is confusing.

Author Response

We appreciate the time and effort taken by the reviewer and have incorporated the suggested changes.

Reviewer 2 comments:

In this review, Leveille and Estall summarize functional roles of mitochondria in carcinogenesis of HCC. This review covers well for mitochondrial dysfunction in NASH and I do not have any comments for this manuscript. However, it would be useful for readers if the authors provide 1-2 figures to represent pathways/associations because there are so many genes and pathways are shown in this manuscript, which is confusing.

Response: We have now added two figures to clarify our idea and summarize the main pathways we have focused on.

We thank the reviewers for their helpful and constructive comments, which we feel have significantly improved the manuscript. We hope that these revisions are sufficient to warrant publishing of the review article.

Round 2

Reviewer 1 Report

I feel the authors have adequately addressed all the comments. This reviewer do not have any more comments to add.